# The Warps and Wefts of a Polyploidy Complex: Integrative Species Delimitation of the Diploid *Leucanthemum* (Compositae, Anthemideae) Representatives

**DOI:** 10.3390/plants11141878

**Published:** 2022-07-19

**Authors:** Tankred Ott, Maximilian Schall, Robert Vogt, Christoph Oberprieler

**Affiliations:** 1Evolutionary and Systematic Botany Group, Institute of Plant Biology, University of Regensburg, D-93053 Regensburg, Germany; maximilian.schall@ur.de (M.S.); christoph.oberprieler@ur.de (C.O.); 2Botanic Garden & Botanical Museum Berlin, Freie Universität Berlin, D-14191 Berlin, Germany; r.vogt@bo.berlin

**Keywords:** consensus K-means, ddRAD, ecological niche modeling, geometric morphometry, integrative taxonomy, *Leucanthemum*, species delimitation

## Abstract

Species delimitation—owing to the paramount role of the species rank in evolutionary, ecological, and nature conservation studies—is an essential contribution of taxonomy to biodiversity research. In an ‘integrative taxonomy’ approach to species delimitation on the diploid level, we searched for evolutionary significant units (the warps and wefts) that gave rise to the polyploid complex of European ox-eye daisies (*Leucanthemum*; Compositae-Anthemideae). Species discovery and validation methods based on genetic, ecological, geographical, and morphometric datasets were applied to test the currently accepted diploid morpho-species, i.e., morphologically delimited species, in *Leucanthemum.* Novel approaches were taken in the analyses of RADseq data (consensus clustering), morphometrics of reconstructed leaf silhouettes from digitized herbarium specimens, and quantification of species-distribution overlaps. We show that 17 of the 20 *Leucanthemum* morpho-species are supported by genetic evidence. The taxonomic rank of the remaining three morpho-species was resolved by combining genealogic, ecologic, geographic, and morphologic data in the framework of von Wettstein’s morpho-geographical species concept. We herewith provide a methodological pipeline for the species delimitation in an ‘integrative taxonomy’ fashion using sources of evidence from genealogical, morphological, ecological, and geographical data in the philosophy of De Queiroz’s “Unified Species Concept”.

## 1. Introduction

Species delimitation, the fundamental rank of taxonomy, is necessary to facilitate ecological, evolutionary, or nature conservation studies. However, of the many species concepts proposed and used in taxonomic studies [1], the majority are not applicable throughout the realm of organismic diversity. Some major reasons for this issue are that speciation is a continuous process [2] and that the relative importance of the different criteria stressed in the different species concepts is variable throughout the tree of life. As a convincing solution to this problem, De Queiroz [3] proposed his “Unified Species Concept” circumscribing species as independently evolving metapopulation lineages, stating that the properties used in other species concepts to define those entities may only be used as indicators for these independently evolving metapopulation lineages. Therefore, these properties (morphology, physiology, ecology, reproductive isolation, geography, etc.) are not helpful in species conceptualization but can be used in species delimitation, and in their conjoint realization add trustworthiness to a delimitation hypothesis.

This conceptual breakthrough paved the way for revitalizing the “biosystematics” approaches to species delimitation from the 1960s and 1970s as “integrative taxonomy” by Dayrat [4] and [5] who questioned the sole DNA-based approaches to species delimitation championed by others (DNA barcoding; [6]). Subsequent research studies have conceptualized this multisource approach to species delimitation and proposed procedural protocols for the joint and/or sequential application of different sources of evidence [7,8] (see [9] for a comparable approach from the “biosystematics” era) and even developed tools for the computationally combined analysis of different lineage properties. As examples of the latter, the joint analyses of morphology, genetics, and geography (Geneland) [10], morphology and geography (“multivariate normal mixtures and tolerance regions” analysis) [11,12], genealogy and morphology (iBPP) [13], or genetics and geography (regression analysis) [14] are of interest.

A general feature of most of the mentioned approaches is that congruent datasets are necessary. This requirement conflicts with the more prevalent situations encountered by taxonomists, in which data for the respective fields of evidence are fragmentary and only partially overlapping. Specimens sampled and genetically analyzed usually represent a subset of specimens from natural collections available for morphological analyses and/or distribution mapping and ecological niche modeling. As a consequence, an integrative species delimitation procedure should be based on a methodological pipeline, in which all fields of evidence are represented by datasets as comprehensive as possible, and which follows a reproducible protocol that allows for the integration of all results for a taxonomical decision. A flexible approach based on the procedures proposed by Schlick-Steiner et al. [7] and Padial et al. [8] will also allow for easier integration of novel or alternative analytical methods from the different fields of evidence, compared to a strict, computationally fixed pipeline. Here, we will apply such an approach to species delimitation questions in a plant group with a complex taxonomical history, namely the diploid representatives of *Leucanthemum* Mill.

The genus *Leucanthemum* Mill. (Compositae, Anthemideae) is a large polyploid complex comprising 42 species [15] with ploidy levels ranging from diploid (2*x*) to dodecaploid (12*x*), and one species [*L. lacustre* (Brot.) Samp.] even showing a chromosome number of 2*n* = 22*x* = 198. The genus is distributed all over the European continent, with one species (the tetraploid *L. ircutianum* DC.) reaching Siberia and some species introduced to many temperate regions of the northern and southern hemispheres [16]. Most of the species are delimited based on differences in morphology, especially leaf morphology, geographical distribution, and ploidy level (e.g., [17,18]). Phylogenetic relationships among the diploids of the genus—the fundamental layer or the “warps and wefts” of the polyploid complex—were addressed in previous studies based on molecular markers (i.e., nrDNA ETS, AFLP fingerprinting, single-copy nuclear markers) [19,20,21,22]. Wagner et al. [22] presented a multi-locus phylogenetic reconstruction of the subtribe Leucantheminae, in which the diversification among diploid *Leucanthemum* species was dated to the last 1.93 (1.14–2.94, 95% credibility interval) Ma, arguing for the strong influence of Pleistocene oscillations on species formation.

Subjects of the present study are the 20 diploid *Leucanthemum* species of Central and Mediterranean Europe. The current species delimitation in this group is largely based on morphological characters, especially leaf shape [17,18]. Instead of delimiting species de novo, we are testing the plausibility of the current almost exclusively morphology-based species delimitation (“morpho-species”), drawing evidence from genetic, morphometric, ecologic, and geographic analyses in the framework of integrative taxonomy. In this context, we present the application of modern methods for the analysis of genetic evidence in the form of biologically informed species delimitation methods and machine learning techniques, morphological evidence by applying methods of leaf reconstruction and geometric morphometry, ecological evidence via ecological niche modeling and niche comparisons, and geographic patterns. By resolving the species boundaries in the diploid *Leucanthemum* representatives, we are laying the foundation for a well-settled taxonomic treatment for this group as a prerequisite for future research on the origin and phylogeny of polyploid representatives of the genus.

## 2. Results

### 2.1. RADseq Assembly

The number of demultiplexed, adapter-clipped, and restriction enzyme filtered reads was 85,612,720 (33.52 mean quality) and 6,936,593 (34.57 mean quality) for the first and second ddRAD batch, respectively. Considering the de novo assembly, optimization of the proposed cost function led to an optimal parameter combination of *ct* = 95 and *msl* = 12 (Appendix A). The corresponding assembly comprised a total of 10,246 RAD loci with 136,755 SNPs, 70,755 of which were parsimony informative. The mean locus coverage per sample was 5035.8 (sd: 795.3). The reference mapping of all 54 *Leucanthemum* accessions against the pseudo-reference resulted in 9248 RAD loci with 155,294 SNPs; 86,390 of which were parsimony informative, at a mean locus coverage of 5415.1 (sd: 721.8) per accession.

### 2.2. Network Analysis

The NeighborNet analysis clustered members from 18 of the 20 *Leucanthemum* morpho-species (Figure 1). For the remaining two species (i.e., *L. cacuminis* and *L. ageratifolium*), a single accession lied outside the cluster containing the remaining individuals of the taxon. There were two larger groups visible in the NeighborNet, the left of them comprising nine morpho-species clusters that were separated from each other by relatively long branches when compared to the right group. In the following, the right cluster will be referred to as *L. vulgare*-group (Figure 1). Within the *L. vulgare*-group, there are two relatively tight clusters around *L. vulgare* and *L. pluriflorum*. We will refer to the first one, comprising *L. vulgare*, *L. pyrenaicum*, and *L. gaudinii* as the *L. vulgare*-cluster, and to the second one, consisting of *L. pluriflorum*, *L. cacuminis*, and *L. gallaecicum*, as the *L. pluriflorum*-cluster. The (nucleotide) Nei distance-based network showed the same clustering (Appendix A).

### 2.3. Hybrid Detection

In total, 5154 ABBA-BABA tests were conducted to detect hybrid individuals, 29 of which were significant (alpha = 0.01 after Bonferroni correction; see Appendix A). The accession M02-01 (*L. ageratifolium*) was involved in 27 of those tests, indicating a strong hybridization signal. The remaining two significant tests involved the triplet consisting of accessions 60-01 (*L. cacuminis*), 68-04 (*L. cacuminis*), and 55-01 (*L. pluriflorum*). Of these individuals, only 68-04 was already noticeable in previous analyses, suggesting a hybrid origin of 68-04.

### 2.4. Species Discovery: Consensus Clustering

According to the Davies–Bouldin (DB) and Silhouette (SIL) indices, the optimal number of clusters for the complete dataset was 15, both with and without hybrid individuals. The consensus clustering with *k* = 15 delimits 13 of the 20 diploid *Leucanthemum* morpho-species (Figure 2A). The remaining seven species form two clusters, one consisting of *L. vulgare*, *L. pyrenaicum*, and *L. gaudinii*, the other comprising *L. pluriflorum*, *L. cacuminis*, *L. gallaecicum*, and *L. eliasii*. The optimal number of clusters for the reduced dataset, i.e., the dataset consisting of the *L. vulgare*-group (see network analysis; Figure 1, right cluster), with and without hybrids was 8 according to SIL, and 11 according to DB. It is noteworthy that there is a local optimum in the DB measures at *k* = 8 (Figure 2B). The consensus clustering with *k* = 8 results in the merging of *L. vulgare* with *L. pyrenaicum* on the one side, and *L. pluriflorum*, *L. cacuminis*, and *L. gallaecicum* on the other. PCoA results are available in Appendix A.

### 2.5. Species Validation: Coalescent-Based Species Delimitation

Marginal likelihood (ML) values were found to increase with the model complexity over the ten scenarios. The scenario with the highest ML was the one, in which each morpho-species formed its own cluster (Figure 3, scenario 10). As the tendency of MSC-based species delimitation methods to overestimate the number of species is known (e.g., [23,24]), we determined the knee of the ML-model complexity curve using the Kneedle algorithm [25], in which we assumed the model complexity being represented by the species number. The knee (or elbow) is the point where the curvature of a function changes considerably; this can be interpreted as the point where increasing the number of species only leads to a relatively low increase in the model performance. We found a knee point at the model with seven groups (scenario 4), which merged *L. vulgare*, *L. gaudinii*, and *L. pyrenaicum*, and *L. pluriflorum*, *L. cacuminis*, and *L. gallaecicum*, respectively, to single groups, while the remaining morpho-species stayed separate (Figure 3 and Appendix A).

### 2.6. Ecological Analysis

The first three principal components (PC) captured 82% and 77% of the total variance considering the BioClim and SoilGrids raster principal coordinate analyses (PCA), respectively. All niche reconstructions using MaxEnt achieved cross-validation AUCs higher than 80%. All pairwise niche-equivalency tests, except for the pair *L. gaudinii*–*L. pyrenaicum*, showed environmental niches being significantly different (Table 1).

### 2.7. Morphological Analysis

Within the *L. vulgare*-cluster, the difference in the PC space based on the descriptors of the elliptic Fourier analysis (EFA) between *L. vulgare* and *L. pyrenaicum* was not significant (corr. *p* = 5.02), while the differences of the combinations *L. vulgare*—*L. gaudinii* (corr. *p* = 0.00) and *L. gaudinii*—*L. pyrenaicum* (corr. *p* = 0.00) were both significant. Within the *L. pluriflorum*-cluster, the test of *L. pluriflorum* against *L.*
*cacuminis* was not significant (corr. *p* = 0.28), while *L. pluriflorum* vs. *L. gallaecicum* (corr. *p* = 0.00) and *L. gallaecicum* vs. *L. cacuminis* (corr. *p* = 0.00) were both significant (Table 1). Considering the leaf dissection index (LDI), the Welch’s tests indicated significant differences (corr. *p* < 0.01) for the two comparisons *L. vulgare* vs. *L. gaudinii*, and *L. pyrenaicum* vs. *L. gaudinii*, while for the remaining comparisons no significant differences could be detected (Table 1).

### 2.8. Geography

In the *L. vulgare*-cluster, the permutation-based tests for geographic overlap indicated a significant allopatry signal for the combination *L. pyrenaicum* vs. *L. gaudinii* (corr. *p* = 0.00), while *L. vulgare* vs. *L. pyrenaicum* (corr. *p* = 6.00) and *L. vulgare* vs. *L. gaudinii* (corr. *p* = 2.22) were not significantly allopatric. In the *L. pluriflorum*-cluster, all tests were significant (Table 1).

## 3. Discussion

With the present study, we aimed at an objective delimitation of the diploid species of the genus *Leucanthemum*—the fundamental layer or the “warps and wefts” of the polyploidy complex this genus forms in central and southern Europe. In its taxonomic scope, the strategy of our analyses followed the operational sequence of (a) grouping and (b) ranking as described by Stuessy [26,27] or Reydon and Kurz [28]. While the former procedure is engaged with finding discontinuities among representatives of a study group in datasets of a different nature (e.g., morphology, genetics, ecology, etc.), the latter evaluates these taxonomic patterns and decides “at which level of the taxonomic hierarchy the taxa should be placed” [26] (p. 623). Species delimitation (SD) frameworks are focused on the taxonomic rank of species and are considered comprising both “discovery” and “validation” approaches [29,30,31]. The “discovery” methods are actually discontinuity-inferring, pattern-recognition techniques that produce hypotheses on a taxonomic structure in the organism group under study, which are subsequently scrutinized by “validation” methods that incorporate models of variation exhibited at and below the species level and, hence, are based on underlying external assumptions (i.e., species concepts).

As in many modern taxonomic studies in areas of the world with a long tradition of (phyto-)taxonomical exploration, our present species delimitation study in *Leucanthemum* diploids did not start from scratch with an uninformed “grouping” step. In terms of adequate sampling of the morphological and genetic diversities in the study area, it was informed by hypotheses on taxonomic structures from former revisionary works based mainly on morphological information [15,17,18,32]. This may appear disadvantageous compared to a “total-ignorance” strategy that would prefer random sampling or sampling along transects or grids, but it represents the more frequent state-of-the-art in taxonomic studies nowadays. Though we have used this morpho-taxon concept for an informed sampling of our accessions in the multilocus RADseq analyses, we have subjected these genomic data both to “discovery” (consensus K-means clustering) and “validation” (coalescent-based species delimitation with SNAPP) analyses. With this morpho-taxa-as-species-hypotheses approach, it was also possible to link the genealogical analyses with geographical, morphological, and ecological datasets based on a larger set of accessions for the final discussion on the ranking of the discovered entities.

We found that the accessions from the morphologically well-characterized species [17,18] outside the *L. vulgare*-group (i.e., *L. gracilicaule*, *L. graminifolium*, *L. rotundifolium*, *L. lithopolitanicum*, *L. halleri*, *L. laciniatum*, *L. tridactylites*, *L. burnatii*, *L. virgatum*) formed genetically distinct and pure groups (Figure 1 and Figure 2A), arguing for their acknowledgment as independent species. Additionally, within the L. vulgare-group, the four morpho-taxa *L. ligusticum*, *L. legraeanum*, *L. monspeliense*, and *L. ageratifolium* were found equally well separated from each other and the *L. vulgare*- and *L. pluriflorum*-clusters, which corroborates a recent species delimitation study in this group based on AFLP-fingerprinting, nuclear DNA markers, and leaf morphology [21]. However, in the remaining closely-knit taxon groups around *L. vulgare* and *L. pluriflorum*, our present “integrative taxonomy” approach has used genealogical, morphological, and ecological “discovery”, and/or “validation” methods for delimitation of entities and additional geographical information for their ranking. For addressing the genealogical, morphological, and geographical layers of evidence, novel methodological approaches were implemented and will be discussed in the following.

*The genealogical layer.* RADseq can be thought of as an approximation of the total genomic variation and is thus well-suited as a source of genetic evidence in the absence of whole-genome sequencing data. We implemented a pseudo-reference-based assembly approach for combining data from multiple ddRAD batches. This pseudo-reference assembly allows for efficiently combining ddRAD datasets while reducing the risk of dataset-specific noise influencing downstream analyses.

Species delimitation based on nucleotide data is known to be subject to the two opposing problems of over- and under-splitting [33]. The former describes the problem of overestimating the number of species by wrongly recognizing the (meta-)population structure as evidence for distinct species. This issue is known to occur especially in the context of species delimitation tools, whose methodology is based on the multispecies coalescent (MSC) model [33] (see [23] for an example). Under-splitting, on the other hand, is the failure of detecting differences between species, consequently merging them into a single one. This problem can occur when hybridization is present because MSC-based species delimitation methods assume the biological species concept (BSC). Under-splitting is a problem that can be relatively easily mitigated by applying hybrid-detection methods to remove those intermediary individuals as we have done with the ABBA-BABA test. However, the problem of over-splitting has not yet been solved when working with the MSC model. Here, we followed the suggestion of Wagner et al. [23] by applying, in addition to a classical MSC-based method (i.e., SNAPP; [34,35]), a non-MSC method (consensus K-means, CKM; [36]) for genetics-based SD. Interestingly, the results of both methods are largely concordant (Figure 2 and Figure 3). Similar observations have been made by Wagner et al. [23]. It might be worthwhile, but out of scope for the present study, to investigate whether this is a general property of the non-parametric and significantly faster CKM to approximate MSC-based SD methods.

We find that 15 of the 20 currently described diploid *Leucanthemum* morpho-species form clusters supported by CKM (Figure 2). Of the remaining five species, the groups consisting of *L. vulgare* and *L. pyrenaicum* (Figure 2B, orange), and *L. pluriflorum*, *L. cacuminis*, and *L. gallaecicum* (Figure 2B, brown) each form a cluster. The MSC-based SD additionally merges *L. gaudinii* with *L. vulgare* and *L. pyrenaicum* (Figure 3). These two clusters, in the following called the *L. vulgare*-cluster and the *L. pluriflorum*-cluster, respectively, were the focal groups of the remaining analyses of ecology, geography, and morphology.

*The morphological layer.* Leaf morphology is a key factor for the currently accepted species delimitation in *Leucanthemum*; in particular, the degree of leaf dissection is considered to be important for distinguishing species [17,18]. The leaf-dissection index (LDI) [37] and elliptic Fourier analysis (EFA) [38] are well-established methods for quantifying variations in leaf shape. While the LDI is invariant if leaves are deformed in such a way that the area and perimeter are conserved, the EFA is sensitive to such distortions. Unger et al. [39] proposed a leaf normalization procedure that aligns pixels of leaf images so that the midvein forms a straight line. However, this method introduces new distortions when the leaves are strongly bent. We picked up the idea by Unger et al. [39] and combined it with a shape manipulation method, which minimizes shape distortions (Figure 4). Future research in this context could focus on the effect of the straightening procedure on the results of geometric morphometric analyses.

Based on the reconstructed leaves, we find that within the *L. vulgare* cluster, the pairs *L. vulgare*–*L. gaudinii* and *L. gaudinii*–*L. pyrenaicum* vary significantly in the degree of leaf dissection and the general leaf shape. Within the *L. pluriflorum* cluster, we could not observe any significant differences in the dissection of the leaves but found that the general leaf shapes of the pairs *L. pluriflorum*–*L. gallaecicum* and *L. gallaecicum*–*L. cacuminis* are significantly different.

*The ecological layer.* Both climatological and edaphic variables were incorporated into our analyses on the abiotic ecological niches of the morpho-species of the two *Leucanthemum* species groups and followed traditional ecological niche modeling (ENM) as proposed by Raxworthy et al. [40]. For niche comparisons, we adopted the niche equivalency tests of Warren et al. [41], whose efficacy for SD has been demonstrated in recent studies (e.g., [42,43]). Our analyses show that the abiotic ecological niches are significantly different for all pairwise comparisons, except for the pair *L. vulgare*–*L. pyrenaicum*. These findings suggest that there is a signal of ecological differentiation even in the absence of clear genetic differentiation, which might indicate young or currently emerging ecotypes.

*The geographical layer.* Even when taxa occupy the same ecological niche, they can still be geographically separated. Conversely, taxa occupying different ecological niches can occur in spatial proximity. Indeed, those patterns have been long used to inform SD [44]. Determining the sympatry of species objectively can be problematic, especially in the absence of distribution maps, as is the case for *Leucanthemum*. Due to this issue, we had to resort to retrieving historical collection data from herbaria. Since herbarium collection data are spatially fragmented and unequally sampled, it is necessary to approximate the true spatial distribution. Established methods for approximating the spatial distribution are convex hulls [45], alpha hulls [45,46,47], or kernel density estimates [41,46] based on collection points. Those methods are very useful when the sampling is mostly complete or when the shape of the spatial distribution follows the corresponding hull function; however, they might fail when the true distribution is only sparsely and spatially disjunctly sampled [46].

Here, we described an ENM-based approach for approximating the true distribution raster from incomplete collection data. This method can be seen as an objective and reproducible variant of a distribution map that a specialist might draw according to the available collection points and expert knowledge of the ecoclimatic variables at play. It might be worthwhile, but out of scope for this contribution, to study how this approximation behaves with other empirical or simulated datasets, and how performant it is at estimating the true distribution area. Based on the approximated raster distribution, we tested the spatial overlap of the species, which we assumed to be a measure of sympatry and allopatry. We found that the currently delimited morpho-species in the *L. pluriflorum* cluster all occur allopatrically and that within the *L. vulgare* cluster only *L. pyrenaicum* and *L. gaudinii* were allopatrically distributed.

*Integration of layers and ranking of entities.* Early formal integration of morphology and geography in species-level taxonomy was proposed by von Wettstein [44]. This author, representing an impressively modern evolutionary approach to taxonomy, considers allopatrically distributed, morphologically similar (i.e., closely-related) units being best-acknowledged at the subspecies level, while species rank should be attributed to closely related, but sympatrically distributed entities. His argument was that only in the latter case, ecological and/or reproductive differentiation between the units is sufficiently advanced to prevent the merging of these lineages. This corresponds to the evolutionary species concept by Wiley [48], who considered species being “ancestor-descendant lineages that evolve separately from other such lineages and have their own evolutionary tendencies and historical fate” [1] (p. 83). The importance of ecological differentiation among species for their geographical co-existence as independent evolutionary lineages is proposed by van Valen [49] in his ecological species concept.

We think that while morphology as a proxy for genetic similarity or genealogical relatedness is nowadays outperformed by genomic data, information on geographical separation and ecological differentiation as drivers, co-variants, and/or consequences of speciation processes definitively should be included in taxonomical decisions concerning ranks at or below the species level. However, morphological information should not be disregarded completely, since discontinuities in this field of taxon properties may still correlate with genealogical differentiation and may either predate, coincide, or follow a speciation event (the Gray Zone of conflicting species concepts in de Queiroz’ [3] (p. 882, Figure 1) argumentation scheme of speciation).

Owing to the pitfalls of an exclusively genealogy-based species delimitation approach, which may lead to the misconception of the population structure for species boundaries [33], we integrated genealogical, morphological, geographical, and ecological data that should cover patterns resulting from a broad array of speciation modes (i.e., allopatric, peripatric, parapatric, or sympatric speciation processes). In the conceptual framework based on von Wettstein’s [44] reasoning, the species level should only be accepted for two lineages that are genealogically distinct if they, simultaneously, are geographically overlapping and (but not necessarily) ecologically distinct. Significant morphological discontinuities between the two lineages may then only allow for the distinction between cryptic and phenetically appreciable species. In line with van Valen’s [49] ecological species concept, allopatrically distributed and genealogically independent lineages should also deserve acknowledgment as species when their ecological niches are different enough to allow for expecting their reproductive isolation in potential sympatry.

*Taxonomical consequences.* With the described conceptual framework at hand, species delimitation in the two *Leucanthemum* clusters under study (the *L. vulgare*-cluster and the *L. pluriflorum*-cluster) could be put into effect as follows:(1)The *Leucanthemum vulgare*-group: While *L. vulgare* is a taxon widespread in Europe, *L. gaudinii* and *L. pyrenaicum* are restricted to the Alps and the Pyrenees, respectively. In genealogical respects, *L. gaudinii* is on the verge of evolving as an independent lineage, with MSC-based SD, in contrast to CKM clustering, showing no significant differentiation, while the other two taxa belong to the same metapopulation system. While both *L. gaudinii* and *L. pyrenaicum* are not significantly allopatric to *L. vulgare*, the former is morphologically different and the latter ecologically. In the case of *L. gaudinii*, its ecological and geographical overlap with *L. vulgare* combined with its genealogical (and morphological) distinctness argues for acknowledgment as an independently evolving lineage (i.e., species), while *L. pyrenaicum* represents only an ecologically deviating facies of *L. vulgare*—an ecotype that could be at best acknowledged taxonomically as a subspecies of the widespread taxon. Consequently, we propose the following taxonomic treatment for this group:(a)***Leucanthemum gaudinii*** Dalla Torre in Sonklar & al., Anleit. Wiss. Beob. Alpenreisen 2: 244. 1882 ≡*Chrysanthemum gaudinii* (Dalla Torre) Dalla Torre & Sarnth., Fl. Tirol 6(3): 543. 1912 ≡*Chrysanthemum leucanthemum* var. *gaudinii* (Dalla Torre) Fiori, Nuov. Fl. Italia 2(4): 624. 1927 ≡*Chrysanthemum leucanthemum* [“f”] *gaudinii* (Dalla Torre) Fiori in Fiori & Paoletti, Fl. Italia 3(1): 239. 1903—Neotypus [Gutermann, Phyton (Austria) 17: 37. 1975]: Kärnten: Nockgebiet, am Weg zur Falkerthütte—Bocksattel—S des Mallnock, 1850 m; mit *Calluna*; flachgründiger Boden über Silikat; leg. *A. Polatschek* P64/312; 2*n* = 18 (W! [W1965-0020139]).(b)***Leucanthemum vulgare*** Lam. [**subsp. *vulgare***], Fl. Franç. 2: 137. 1779 ≡*Chrysanthemum leucanthemum* L., Sp. pl. 888. 1753 ≡*Tanacetum leucanthemum* (L.) Sch.Bip., Tanaceteen: 35. 1844 ≡*Matricaria leucanthemum* (L.) Desr. in Lam., Encycl. 3(2): 731. 1792—Lectotypus [Böcher & Larsen, Watsonia 4: 15. 1957]: (BM-Hortus Cliffortianus).(c)***Leucanthemum vulgare* subsp. *barrelieri*** (Dufour ex DC.) O.Bolós & Vigo, Pl. Països Catalans 3: 816. 1996 “1995” ≡*Pyrethrum halleri* var. *barrelieri* Dufour ex DC., Prodr. 6: 55. 1838 (basionym) ≡*Leucanthemum gaudinii* subsp. *barrelieri* (Dufour ex DC.) Vogt in Ruizia 10: 89. 1991 ≡*Leucanthemum ceratophylloides* var. *barrelieri* (Dufour ex DC.) Nyman, Consp. Fl. Eur.: 371. 1879 ≡*Pontia barrelieri* (Dufour ex DC.) Bubani, Fl. Pyr. 2: 219. 1899 ≡*Pyrethrum barrelieri* Dufour ex DC., Prodr. 6: 55. 1838, pro. syn., nom. inval. ≡*Leucanthemum vulgare* var. *pyrenaicum* Rouy, Fl. France 8: 272 and 274. 1903, nom. illegit. ≡*Leucanthemum pyrenaicum* Rouy, Fl. France 8: 274. 1903, pro syn., nom inval. [non *Leucanthemum barrelieri* Timb.-Lagr., Bull. Soc. Bot. France 13: 153. 1866 and in Rodet, Bot. Agric. Médic., ed. 2: 447. 1872] ≡*Leucanthemum pyrenaicum* Vogt et al. in Mol. Phylogenet. Evol. 92: 325. 2015.—Holotype: Pyren., près du Sommet de Monné, 1824 (G-DC! [G00450856]).(2)The *Leucanthemum pluriflorum*-group: All three morpho-species of this group are allopatrically distributed: while *L. pluriflorum* is restricted to the coastline of Galicia in NW Spain (see [17,18,50]), *L. gallaecicum* is endemic to serpentine outcrops in central Galicia [51], and *L. cacuminis* (the former *L. gaudinii* subsp. *cantabricum**;* [20]) is found in the mountainous regions of Northern Spain from the western Pyrenees in the east to the Picos de Europa in the west. Since no significant genealogical independence among the three taxa was found, acknowledgment on the species level is questionable, despite the significant non-overlapping in ecological respects of all three taxa and the morphological distinction of *L. gallaecicum* from the other two taxa. Therefore, our interpretation of the *L. pluriflorum*-group as a lineage of allopatrically and ecologically differentiating population groups may be best represented taxonomically by ranking the three taxa as subspecies of a single species:(a)***Leucanthemum pluriflorum*** Pau [**subsp. *pluriflorum***] in Bol. Soc. Aragonesa Ci. Nat. 1: 31. 1902—Holotype: San Ciprián, Galicia, *P. Merino S.J.* (MA! [MA128479]).(b)***Leucanthemum pluriflorum*****subsp. *cantabricum*** (Font Quer & Guinea) T.Ott, Vogt & Oberpr., **comb. nov.** ≡*Leucanthemum vulgare* var. *cantabricum* Font Quer & Guinea in Guinea, Anales Jard. Bot. Madrid 7: 347–348. 1947 (basionym) ≡*Chrysanthemum leucanthemum* subsp. *cantabricum* (Font Quer &Guinea) Guinea, Catálogo florístico de Viscaya: 646. 1980 ≡*Leucanthemum gaudinii* subsp. *cantabricum* (Font Quer & Guinea) Vogt in Ruizia 10: 98. 1991 ≡*Chrysanthemum leucanthemum* var. *cacuminis* Font Quer & Guinea in Guinea, Bot. Santander: 327. 1953, nom. inval. ≡*Leucanthemum cacuminis* Vogt et al. in Mol. Phylogenet. Evol. 92: 325. 2015.—Holotype: Picos de Europa: in saxosis l. Vega de Liordes, ad 1890 m alt., 13.8.1944*, E. Guinea* (BC!).(c)***Leucanthemum pluriflorum* subsp. *gallaecicum*** (Rodr.Oubiña & S.Ortiz) T.Ott, Vogt & Oberpr., **comb. et stat. nov.** ≡*Leucanthemum gallaecicum* Rodr.Oubiña & S.Ortiz in Anales Jard. Bot. Madrid 47: 498. 1990 (basionym).—Holotype: La Coruña: Toques, Paradela, 20-IX-1987, *J. Rodríguez Oubiña* & *S. Ortiz* (SANT; isotype: MA! [MA478919]).

## 4. Materials and Methods

### 4.1. RADseq Assembly

For the Double Digest RADseq (ddRADseq) [52] procedure, a set of 54 individuals representing the 20 diploid *Leucanthemum* morpho-species was selected (see Table 2), comprising at least 2 individuals per taxon. The ddRAD data were generated in two batches, the first one comprising 52 silica-dried leaf samples from all morpho-species except *L. eliasii*, the second one 2 samples of *L. eliasii* from herbarium specimens. Genomic DNA was extracted according to the CTAB DNA extraction protocol of [53]. Two replicates were generated by repeating the extraction process for *L. ageratifolium* (M60-01, M60-011) and *L. halleri* (162-03, 162-031). The DNA extracts were forwarded to LGC Genomics (Berlin, Germany) for ddRAD (2 × 150 bp) Illumina sequencing on an Illumina NextSeq 500 instrument (Illumina, Inc., San Diego, CA, USA) with the restriction enzyme combination *Pst1* and *ApeK1*.

The raw reads were demultiplexed and adapter-clipped by LGC Genomics. Since there is no reference genome available for *Leucanthemum*, de novo assembly of the preprocessed reads of the first ddRAD batch was performed using iPyrad version 0.9.54 [54]. The pipeline parameters determining the clustering threshold (clust_threshold, *ct*) and the minimum number of samples required for a single locus to be retained (min_samples_locus, *msl*) were optimized, while the remaining parameters were kept at the default values. For the optimization, a cost function accounting for the assembly error of in vitro replicates and the amount of missing data in the assembly were applied. The cost function is described by the formula $\left(1-E_{L}\right)*\left(1-E_{S}\right)*g\left(M_{S},\;\mu =50.0,\;\sigma =50.0\right)$, where $E_{L}$ is the (mean) locus error rate, and $E_{S}$ the (mean) single nucleotide polymorphism (SNP) error rate considering the in vitro replicates, $M_{S}$ is the percentage of missing data in the SNP data matrix returned by iPyrad, and $g\left(M_{S},\;\mu ,\;\sigma \right)$ is the Gaussian transformation of $M_{S}$ with mean μ and standard deviation σ. The replicate-based error measures $E_{L}$ and $E_{S}$ were calculated as described by Mastretta-Yanes et al. [55]. The idea behind this cost function was to combine the replicate-based error measures $E_{L}$ and $E_{S}$ with a rough target for the allowed amount of missing data in the assembly $g\left(M_{S},\;\mu ,\;\sigma \right)$. Assemblies with all parameter combinations of *msl* ranging from 4 to 52 in increments of 4 and *ct* ranging from 85 to 95 in increments of 2 were constructed. The assembly scoring best according to the previously described cost function was selected for constructing a ddRAD pseudoreference.

The aligned locus sequences were used to calculate a majority consensus sequence for each locus. The locus consensi were used as the pseudoreference for a subsequent reference-based assembly using iPyrad of all 54 *Leucanthemum* accessions, keeping all parameters at default, except for *msl*, which was set to the optimal value according to the optimization procedure.

### 4.2. Network Analysis

As a first exploratory analysis, the NeighborNets were calculated using SplitsTree4 version 4.15.1 [56] based on Kimura-2-parameter (K2P) distances of the concatenated SNPs (iPyrad’s. snps. phy output) and concatenated loci, and on SNP-based Nei distances calculated from the variant information (.vcf output). K2P distances with pairwise-deletion were calculated using the R package ape version 5.4 [57]; for calculation of the Nei distances, a custom Python and C implementation of the Nei distance on the SNP level [58] as described in the POFAD manual [59,60] was employed.

### 4.3. Hybrid Detection

Hybrid detection in the reduced dataset was performed using Patterson’s D-statistics (“ABBA-BABA tests”) [61,62] implemented in iPyrad, as described by Wagner et al. [23]. For each possible pair of morpho-species, all possible pairs of individuals from those species were subjected to Bonferroni-corrected ABBA-BABA tests.

### 4.4. Consensus Clustering

The principal coordinate analysis (PCoA) was applied to transform pairwise K2P distances of the concatenated SNPs into a principal coordinates (PCo) matrix. This was conducted for the whole dataset and for a reduced dataset comprising the samples from eleven morpho-species (*L. vulgare*-group) that were found being less clearly delimited according to the previous NeighborNet analysis (see Section 2.2, Figure 1). For both datasets, only the first PCos explaining at least 80% of the total variance were retained. Based on the PCo matrices, consensus K-means (CKM) [36] clustering was conducted as described by [23]. The clusters *k* numbers varied from 2 to 20 and 2 to 11, for the complete and reduced datasets, respectively. For both, CKMs were calculated with and without hybrid individuals. The feature and observation portions were both set to 0.8, meaning that the single K-means runs (replicates) within a CKM use a random subset of the data matrix containing 80% of the samples and features. The number of replicates was set to 5000. For both CKM analyses, the best *k* was determined using the Davies–Bouldin Index and the silhouette index. The analyses were conducted using the Python package pyckmeans version 0.9.4 [63].

### 4.5. Coalescent-Based Species Delimitation

Multispecies coalescent (MSC) species delimitation was performed using the BEAST2 version 2.5.2 [64] package SNAPP version 1.4.2 [34,35] based on the reduced dataset without putative hybrid individuals. Ten different species membership scenarios (S1–S10, see Figure 3) were surveyed by grouping taxa into clusters that were reasonable according to the current taxonomy and to previous analyses. SNAPP input files were constructed using BEAUTI version 2.5.2 [64]. Since SNAPP analyses are computationally expensive, it was not possible to use the full SNP alignment (.phy.snps). Instead, the iPyrad’s “.phy.usnps” output, comprising one randomly drawn SNP per locus, was used. The prior for the Yule model (lambda) was set to a gamma distribution with *alpha* = 2 and *beta* = 200. The population size prior was set to a gamma distribution with *alpha* = 1 and *beta* = 250. These parameter choices were motivated by the tutorial of Leaché and Bouckaert [65]. MCMC sampling (500,000 iterations, 500 sampling rate, 25% burn-in, *alpha* = 0.3) for the 10 scenarios with 48 path-sampling steps each, was conducted on the Athene HPC cluster at the University of Regensburg.

### 4.6. Ecological Niche Modeling

Ecological niche modeling (ENM) was used to reconstruct potential distribution areas and compare the eco-climatological and edaphic niches of the six *Leucanthemum* morpho-species (Section 2.2, Figure 1; *L. vulgare*-cluster: *L. vulgare*, *L. gaudinii*, *L. pyrenaicum*; *L. pluriflorum* cluster: *L. pluriflorum*, *L. cacuminis*, *L. gallaecicum*), which could not be reliably delimited using genetic data alone (see Section 2.2, Section 2.3, Section 2.4 and Section 2.5, Figure 1 and Figure 2). A total of 732 collection locations of individuals from these taxa were retrieved from several herbaria (ARAN, B, BC, BCC, BCN, COI, E, FR, G, GOET, JACA, JBAG, LEB, LOU, LY, M, MA, MAF, MGC, NEU, P, PAD, SALA, SANT, SEV, TSM, VAL, VIT, W, WU; abbreviations according to the Index Herbariorum [66]; see Appendix A).

Rasters of 19 bioclimatic variables and ten edaphic variables from three different soil depth levels (0–5 cm, 5–15 cm, 15–30 cm) were retrieved from Worldclim Bioclim [67] and SoilGrids [68], respectively (see Appendix A for variable descriptions). The rasters were cropped to a bounding box enclosing central Europe (longitude: −10.0–26.0°; latitude: 35.9–52°) and scaled to the resolution of the Bioclim rasters (2.5 min) using the R package raster version 3.3 [69]. The three depth levels of each of the edaphic variables were summarized by calculating the mean cell values, resulting in ten edaphic rasters. The principal component analysis (PCA) with normalization implemented in the R package ENMtools version 1.0.2 [70] was applied to the Bioclim and SoilGrids rasters for dimensionality reduction and decorrelation. For each dataset, the three principal component rasters explaining the largest part of variance in the dataset were selected for subsequent analyses.

Based on the six principal component rasters and 732 collection locations, potential distribution areas were reconstructed for each of the six morpho-species using MaxEnt version 3.4.4 [71]. The cross-validation option was set to 6-fold. Niche equivalency, implemented in the R package ENMTools version 1.0.6 [70], was inferred among the morpho-species using MaxEnt as ENM, a species range of 50 km, 1000 background points with a radius of 20 km, and 200 permutations.

### 4.7. Morphological Analyses

For the morphological analyses of six genealogically close morpho-species (*L. vulgare*-cluster, *L. pluriflorum*-cluster), 66 images of digitized herbarium specimens were provided by the herbarium of the Berlin Botanical Museum (B; see Appendix A). Based on the digital images, largely intact, intermediate leaves were manually segmented (annotated) with polygons, and the corresponding midveins were annotated with polylines, using the Computer Vision Annotation Tool (CVAT) software version 3.8.0 [72]. In total, 417 contour-midvein pairs (i.e., leaves) were retrieved; annotated images were exported as the CVAT image (XML) dataset.

Leaf deformations that very likely influence the results of downstream geometric morphometric analyses were compensated by straightening the leaves using a custom procedure implemented in Python and C based on the as-rigid-as-possible algorithm [73] and relying on functionality from the Python packages scikit-image [74], numpy [75], triangle [76,77], and igl [78]. The process comprises four steps: (1) generate a triangle mesh from the leaf polygon and midvein polyline by applying a constrained Delaunay triangulation [79], (2) move the midvein vertices in such a way that they form a straight line while preserving distances between the vertices, (3) move all other vertices according to the as-rigid-as-possible algorithm [73], and (4) map the leaf texture from the original triangle mesh to the transformed triangle mesh (Figure 4).

To extract morphological features from the straightened leaf images, the elliptic Fourier analysis (EFA) [38] was conducted and leaf dissection indices (LDI) [37] were calculated. LDI calculation and EFA were performed with the Python packages scikit-image [74], numpy [75], scikit-learn [80] and pyefd [81] For the EFA, the number of harmonics was set to 15 and the elliptic Fourier descriptors (EFDs) were normalized. The non-constant EFDs were subjected to a principal component analysis (PCA) for decorrelation.

The first two principal component (PC) scores were used for permutation-based testing for significant differences in leaf shapes among morpho-species. For all possible pairs of morphospecies within genealogical clusters, the observed Euclidean distances among individuals in the PC space were calculated. The observed distances were tested for significance against simulated null distributions generated by randomly swapping taxon labels and calculating the corresponding distances 5000 times. The *p*-values were Bonferroni-corrected. Additionally, the LDI values were tested for significant differences using Bonferroni-corrected Welch’s tests.

### 4.8. Geography

Pair-wise geographic overlap between taxa is an important factor for species delimitation in our integrative taxonomical approach. Since there are no complete collection maps available for the six genealogically weakly delimited *Leucanthemum* morpho-species (*L. vulgare*-cluster, *L. pluriflorum*-cluster), it was necessary to approximate the true geographic distribution from the incomplete sampling data available. For this purpose, an ENM was fitted using MaxEnt [71], based on the BioClim, SoilGrid, and, additionally, longitude and latitude rasters. For each morpho-species, the predicted suitability raster was subjected to a thresholded depth-first search (DFS), originating from those raster cells, for which collection data were available. A cell was only visited by the DFS if the corresponding suitability score was above a defined threshold, in this case, 0.25; cells that were not visited were set to 0.0. Finally, cells containing collection points were set to 1.0. The thresholded DFS removes areas that are disconnected from true sampling locations and additionally filters the suitability rasters according to the threshold. In the following, those filtered rasters were treated as approximate species-distribution maps.

Based on these, the pairwise overlap between morpho-species was calculated by (1) multiplying the approximate distribution rasters, (2) thresholding the resulting raster by 0.25, (3) calculating the number of non-zero raster cells, and (4) dividing by the number of non-zero raster cells of the species with the smaller distribution area. To test for significant deviation from sympatry, a permutation-based approach was taken. For each comparison, 400 datasets were simulated by randomly swapping taxon labels, the distribution areas were approximated, and the overlap was calculated. Finally, the *p*-value was calculated by comparing the observed overlap with the simulated overlap. In this setup, a significant deviation from the null model in direction of a lower overlap means a deviation from sympatry that cannot be explained by chance alone.

## Figures and Tables

**Figure 1 plants-11-01878-f001:**
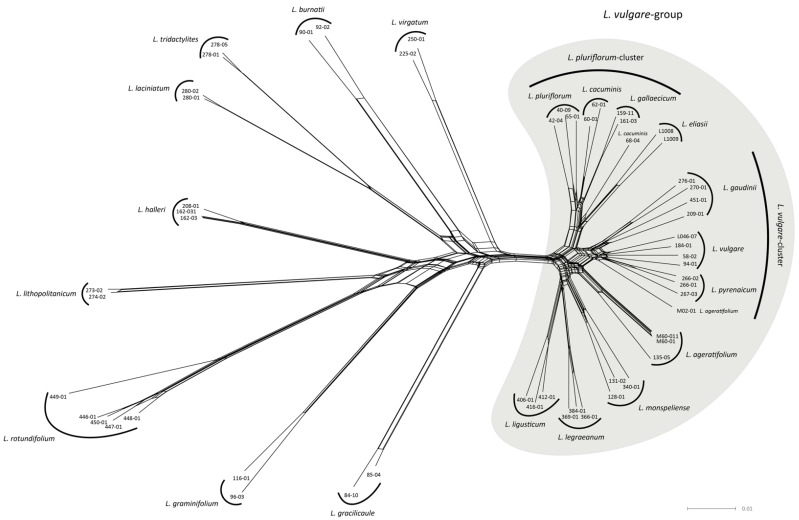
NeighborNet of the 20 *Leucanthemum* morpho-species, constructed using Kimura 2-parameter distances of the concatenated ddRAD loci from the iPyrad pseudo-reference assembly. Species membership is indicated by enclosing curves or text for single accessions. Accessions of 18 species form tight clusters, while accessions M02-01 *(L. ageratifolium)* and 68-04 (*L. cacuminis*) lie outside of the respective species clusters.

**Figure 2 plants-11-01878-f002:**
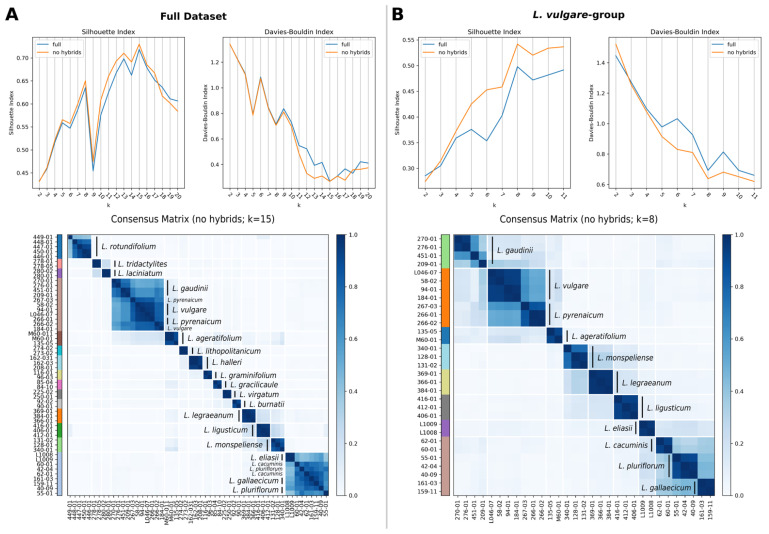
Consensus K-means clustering results for the complete (**A**) and reduced (**B**) datasets. The line charts show the clustering quality metric silhouette index (SIL; higher is better) and Davies–Bouldin index (DB; lower is better) for datasets both with (blue) and without (orange) hybrid individuals (M02-01, 68-04). For the complete dataset, the optimal number of clusters is 15, independent of the presence of hybrid individuals. Considering the reduced dataset, the optimal *k* is either 8 or 11, according to SIL and DB, respectively. For DB, there is a local optimum at *k = 8*. The heat maps depict the consensus matrices for the optimal *k* of the complete (**left**) and reduced (**right**) datasets excluding hybrid individuals.

**Figure 3 plants-11-01878-f003:**
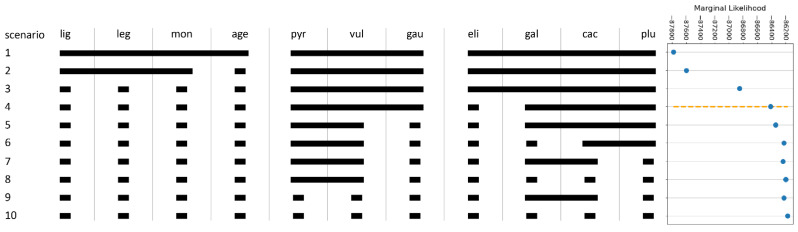
The ten species scenarios (scenarios 1–10) from merging the morpho-species in ascending orders of complexity and the plot of the marginal likelihoods (MLs) against species scenarios. The elbow point of the ML-complexity curve (orange dotted line) is located in scenario 4.

**Figure 4 plants-11-01878-f004:**
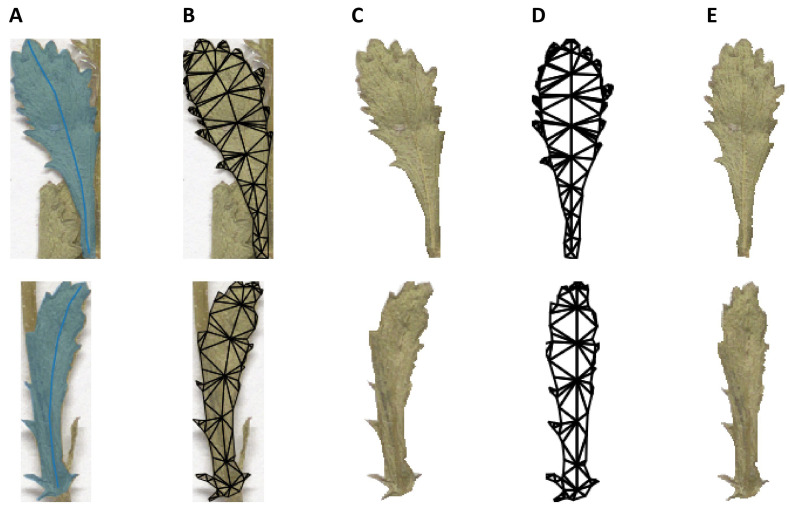
Leaf straightening process illustration. The leaf and midvein annotations (**A**) are used to construct a triangle mesh (**B**). Based on this mesh, the background is removed (**C**). The midvein points are arranged in a straight line, preserving distances between line vertices; the remaining vertices are updated according to the as-rigid-as-possible algorithm (**D**). Finally, the texture from the bent leaf mesh is mapped to the straightened leaf mesh (**E**).

**Table 1 plants-11-01878-t001:** Significance table for the pairwise comparisons in the *L. pluriflorum*- and *L. vulgare*-groups, with *p*-values for the environmental niche-overlap metric *D,* the permutation tests of the elliptic Fourier analysis (*EFA*), the Welch’s tests of the leaf-dissection index (*LDI*), and the permutation test of the geographical overlap (*geo*). In all cases, the *p*-values were Bonferroni-corrected. Corrected *p*-values greater than 1.0 are truncated to 1.0. Bold numbers indicate significant *p*-values.

Taxon A	Taxon B	*p (D)*	*p (EFA)*	*p (LDI)*	*p(geo)*
*L. gaudinii*	*L. vulgare*	**0.03**	**0.00**	**0.02**	1.00
*L. gaudinii*	*L. pyrenaicum*	0.06	**0.00**	**0.00**	**0.00**
*L. vulgare*	*L. pyrenaicum*	**0.03**	1.00	0.34	1.00
*L. pluriflorum*	*L. gallaecicum*	**0.03**	**0.00**	0.67	**0.00**
*L. pluriflorum*	*L. cacuminis*	**0.03**	0.28	1.00	**0.00**
*L. gallaecicum*	*L. cacuminis*	**0.03**	**0.00**	0.42	**0.00**

**Table 2 plants-11-01878-t002:** List of samples used for the ddRAD analysis with information on voucher specimens (in B) and collection localities, coordinates, and collectors.

Sample	Taxon	Voucher Specimens	Locality	Coordinates (Latitude, Longitude)	Collection Number
135-05	*L. ageratifolium* Pau	B100386712	F, Occitania, Pyrénées-Orientales, 410 m	42.5038, 2.9603	Konowalik KK42 and Ogrodowczyk
M60-01	*L. ageratifolium* Pau	B100345012, B100345013	ES, Castile-La Mancha, Cuenca, 1157 m	40.1019, −1.521	Cordel 60
M02-01	*L. ageratifolium* Pau	B100297950	ES, Aragón, Huesca, 755 m	42.525, −0.669	Cordel 2
90-01	*L. burnatii* Briq. and Cavill.	B100464678	F, Provence-Alpes-Côte d’Azur, Alpes-Maritimes, 1235 m	43.7607, 6.9165	Vogt 16615 et al.
92-02	*L. burnatii* Briq. and Cavill.	B100464676, B100464675	F, Provence-Alpes-Côte d’Azur, Bouches-du-Rhône, 650 m	43.545, 5.6626	Vogt 16618 et al.
60-01	*L. cacuminis* Vogt et al.	B100413746	ES, Galicia, Os Ancares, 1530 m	42.8315, −6.8569	Hößl 60
62-01	*L. cacuminis* Vogt et al.	B100413744	ES, Galicia, Lugo, 750 m	42.9249, −6.8657	Hößl 62
68-04	*L. cacuminis* Vogt et al.	B100413738	ES, Cantabria, Cantabria, 1770 m	43.1538, −4.8053	Hößl 68 and Himmelreich
L1008	*L. eliasii* (Sennen and Pau) Sennen and Pau	B100484003	ES, Castile and León, Burgos, 880 m	42.503, −3.706	Lopéz 2537 et al.
L1009	*L. eliasii* (Sennen and Pau) Sennen and Pau	B100484004	ES, Castile and León, Burgos, 920 m	42.507, −3.705	Galán Cela 576 and Martín
159-11	*L. gallaecicum* Rodr. Oubiña and S. Ortiz	B100386789, B100420775, B100464989	ES, Galicia, Pontevedra, 375 m	42.8498, −7.9878	Konowalik KK67 and Ogrodowczyk
161-03	*L. gallaecicum* Rodr. Oubiña and S.Ortiz	No voucher	ES, Galicia, Corunna, 380 m	42.8533, −7.9994	Konowalik s.n. et al.
58-02	*L. gallaecicum* Rodr. Oubiña and S. Ortiz	B100413748	ES, Galicia, Lugo, 490 m	42.8205, −7.9504	Hößl 58
209-01	*L. gaudinii* Dalla Torre	B100386664	CH, Bern, Interlaken-Oberhasli, 2260 m	46.5781, 7.97	Tomasello TS88
270-01	*L. gaudinii* Dalla Torre	B100413007	AT, Carinthia, Spittal an der Drau, 2200 m	47.0025, 13.5275	Oberprieler 10859
276-01	*L. gaudinii* Dalla Torre	B100413015	AT, Carinthia, Feldkirchen, 2270 m	46.8603, 13.8172	Oberprieler 10866
451-01	*L. gaudinii* Dalla Torre	No voucher	PL, Lesser Poland, Giewont, 1860 m	49.2505, 19.9343	Konowalik 20160909-01
84-10	*L. gracilicaule* (Dufour) Pau	B100386704	ES, Valencian Community, Alicante, 300 m	38.8379, −0.1853	Konowalik KK20 and Ogrodowczyk
85-04	*L. gracilicaule* (Dufour) Pau	B100386702	ES, Valencian Community, Valencia, 340 m	39.3135, −0.681	Konowalik KK25 and Ogrodowczyk
116-01	*L. graminifolium* (L.) Lam.	B100464684, B100464683	F, Occitania, Hérault, 800 m	43.7761, 3.2386	Vogt 16693 et al.
96-03	*L. graminifolium* (L.) Lam.	B100464663	F, Occitania, Aude, 600 m	43.1494, 2.6294	Vogt 16656 et al.
162-03	*L. halleri* (Vitman) Ducommun	B100386798	D, Bavaria, Landkreis Garmisch-Partenkirchen, 2340 m	47.4134, 11.1277	Konowalik KK67 and Tomasello
208-01	*L. halleri* (Vitman) Ducommun	B100386672	CH, Valais, Sion, 2320 m	46.3308, 7.2911	Tomasello TS65
280-01	*L. laciniatum* Huter et al.	B100464203	I, Calabria, Cosenza, 1580 m	39.902, 16.1144	Tomasello 420
280-02	*L. laciniatum* Huter et al.	B100464203	I, Calabria, Cosenza, 1580 m	39.902, 16.1144	Tomasello 420
366-01	*L. legraeanum* (Rouy) B.Bock and J.-M.Tison	B100486634, B100486635, B100486636, B100486637, B100486638	F, Provence-Alpes-Cote d’Azur, Var, 410 m	43.1986, 6.3151	Vogt 17189
369-01	*L. legraeanum* (Rouy) B.Bock and J.-M.Tison	B100486648, B100486649	F, Provence-Alpes-Cote d’Azur, Var, 210 m	43.2444, 6.3377	Vogt 17192
384-01	*L. legraeanum* (Rouy) B.Bock and J.-M.Tison	B100627809, B100627810	F, Provence-Alpes-Cote d’Azur, Var, 410 m	43.1988, 6.3151	Vogt 17434 et al.
406-01	*L. ligusticum* Marchetti et al.	B100627838, B100627839	I, Liguria, La Spezia, 210 m	44.247, 9.7728	Vogt 17460 et al.
412-01	*L. ligusticum* Marchetti et al.	B100627849, B100627850, B100627851	I, Liguria, Genova, 700 m	44.3603, 9.5105	Vogt 17468 et al.
416-01	*L. ligusticum* Marchetti et al.	B100627855, B100627856	I, Liguria, Genova, 250 m	44.3458, 9.4588	Vogt 17471 et al.
273-02	*L. lithopolitanicum*	B100413012	SL, Central Slovenia, 2100 m	46.3633, 14.5715	Oberprieler 10862
274-02	*L. lithopolitanicum*	B100413013	SL, Savinja, 2000 m	46.375, 14.5663	Oberprieler 10864
128-01	*L. monspeliense* (E. Mayer) Polatschek	B100464618	F, Occitania, Gard, 750 m	44.0888, 3.5786	Vogt 16712 et al.
131-02	*L. monspeliense* (E. Mayer) Polatschek	B100464615	F, Occitania, Gard, 380 m	44.1412, 3.7316	Vogt 16716 et al.
340-01	*L. monspeliense* (E. Mayer) Polatschek	B100486666, B100486667	F, Occitania, Aveyron, 180 m	44.5822, 2.184	Vogt 17156 et al.
40-09	*L. pluriflorum* Pau	B100413758	ES, Galicia, Corunna, 100 m	42.8838, −9.2726	Hößl 40
42-04	*L. pluriflorum* Pau	No voucher	ES, Galicia, Corunna, 150 m	43.3069, −8.6186	Hößl 42
55-01	*L. pluriflorum* Pau	B100413749	ES, Galicia, Lugo, 10 m	43.6309, −7.333	Hößl 55
266-01	*L. pyrenaicum* Vogt et al.	B100464208	ES, Aragon, Huesca, 1650 m	42.7806, −0.2467	Tomasello TS382
266-02	*L. pyrenaicum* Vogt et al.	B100464208	ES, Aragon, Huesca, 1650 m	42.7806, −0.2467	Tomasello TS382
267-03	*L. pyrenaicum* Vogt et al.	B100464210	ES, Aragon, Huesca, 2000 m	42.6327, 0.453	Tomasello TS392
446-01	*L. rotundifolium* (Willd.) DC.	No voucher	PL, Podkarpackie, Bieszczady, 920 m	49.11905, 22.57755	Konowalik 20180622-02-01
447-01	*L. rotundifolium* (Willd.) DC.	No voucher	RO, Bihor, Bihor, 1230 m	46.51887, 22.66133	Konowalik 20180713-03-01
448-01	*L. rotundifolium* (Willd.) DC.	No voucher	BH, Central Bosnia Canton, 1860 m	43.95782, 17.74027	Konowalik 20180714-03-01
449-01	*L. rotundifolium* (Willd.) DC.	No voucher	RO, Hunedoara, Râu de Mori, 1140 m	45.31588, 22.77045	Konowalik 20180807-03-01
450-01	*L. rotundifolium* (Willd.) DC.	No voucher	PL, Lesser Poland Voivodeship, Sucha County, 1100 m	49.58787, 19.55152	Konowalik 20170920-01
278-01	*L. tridactylites* (A. Kern. and Huter) Huter et al.	B100464207	I, Abruzzo, Pescara, 2080 m	42.1384, 14.1101	Tomasello 417
278-05	*L. tridactylites* (A. Kern. and Huter) Huter et al.	B100464207	I, Abruzzo, Pescara, 2080 m	42.1384, 14.1101	Tomasello 417
225-02	*L. virgatum* (Desr.) Clos	B100411746	F, Provence-Alpes-Côte d’Azur, Alpes-Maritimes, 430 m	43.9538, 7.2961	Vogt 16892 and Oberprieler
250-01	*L. virgatum* (Desr.) Clos	B100350169, B100350172	I, Liguria, Savona, 220 m	44.0596, 8.0583	Vogt 16932 and Oberprieler
184-01	*L. vulgare* Lam.	B100346626	BH, Republic of Srpska, Nevesinje, 930 m	43.2403, 18.3364	Vogt 16806 and Prem-Vogt
L046-07	*L. vulgare* Lam.	B100550249	D, Bavaria, Deuerling, 450 m	49.0333, 11.8833	Eder and Oberprieler s.n.
94-01	*L. vulgare* Lam.	B100464674	F, Occitania, Aude, 160 m	43.1294, 2.6073	Vogt 16641 et al.

## Data Availability

The raw ddRAD reads were deposited at NCBI (Bioproject PRJNA857845).

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
