# Peer review of "The Warps and Wefts of a Polyploidy Complex: Integrative Species Delimitation of the Diploid Leucanthemum (Compositae, Anthemideae) Representatives"

_plants, 2022, doi:10.3390/plants11141878_

Round 1

Reviewer 1 Report

The paper presents an innovative and comprehensive approach for the delimitation of species following an ‘integrative taxonomy’ that exploits multiple sources of evidence, which is exemplified in the group of diploid Leucanthemum species. The study is excellent, well done and presented; it provides significantly new insights into the studied group  and proposes a new methodological procedure for robust and objective species delimitation in taxonomically intricate species complexes.

I see only a few weak points in the manuscript, which should be considered by the authors, as listed below.

Introduction

The aims, research question, or hypotheses tested in the present study should be better formulated, to express them more exactly, to better focus and stress the relevance of the presented study. In the present version of the manuscript, the formulations such as “we are testing the plausibility of the current species delimitation” or “we present the application of modern methods…” appear fairly weak to me, and do not emphasize sufficiently the relevance of the study. I see here also some contradicting statements, such as “Instead of delimiting species de-novo…” vs “By resolving the species boundaries…”. In addition, the authors stated that the current species delimitation is largely based on morphological characters and ecology, but further on in the manuscript the entities delimited and tested are referred to throughout the text as ‘morpho-species’. What do you mean by ‘morpho-species’ – those traditionally delimited and recognized species largely based on morphological (leaf shape) traits? Please explain this term when using it for the first time in the manuscript. Regarding the ecology – so what has already been known about the ecological divergence among those morpho-species, and how is this knowledge expanded by the here presented ENM analyses?

Thus, I think the authors should pay more attention to the last paragraph of the Introduction to make it more clear, better focused and highlight the innovative character of the presented study.

Results

The authors refer to a L. vulgare-group (or -cluster) as late as in p. 4, line 137:
“i.e., the dataset consisting L. vulgare-group (see network analysis),” however, without proper definition or explanation what does this group consist of. Please note that this group is not mentioned in any way when describing the NeighborNet results (neither in the text, nor in the graph itself), so the reference “see network analysis” is confusing. Similarly, the authors refer to L. pluriflorum-cluster in p. 5, line 180, but this cluster or group has not been explained earlier. I strongly suggest to explain and delimit these two groups already in the 2.2 section and indicate them also on Fig. 1. Otherwise the Results section is difficult to follow and some parts can be understood well only after reading the Discussion section. If these two groups somehow refer to the groups of 11 or 6 species already mentioned in the M&M (sections 4.6., 4.7.) these names should already be included there.

As late as in the Discussion (p. 7, lines 264-266), it is specified that these two groups/cluster are in fact the focal groups of the present study. This fact definitely should appear on a much earlier place of the manuscript (I suggest even in the Introduction).

The order of ‘Consensus clustering’ and ‘Hybrid Detection’ paragraphs differs between the M&M and Results sections. In the M&M it is a bit confusing to read “with and without hybrid individuals” in 4.3 paragraph (line 475) when the hybrid detection paragraph (4.4) appears later.

2.6. section, line 174

I suppose, instead of L. vulgare – L. gaudinii it should be L. pyrenaicum – L. gaudinii (following Table 1)

M&M section

4.2. Network analysis

The authors refer to NeighborNet analyses computed both from K2P and Nei distances, but only the former is mentioned in the Results.

4.6., 4.7, 4.8 ENM, morphology, geography

I am missing here a clear explanation why only those 6 species were included here, why not all 20 or 11 (=a reduced dataset, mentioned above)? Are those 6 species a subset of those 11 ones? I understand that they refer to those L. vulgare- and L. pluriflorum-clusters, but it becomes clear only after reading the Discussion section.

Discussion

I lack any discussion on the detected hybridization - its extent, ecological and geographic circumstances (occuring in area overlaps?), consequences for species or subspecies delimitations, etc. I am wondering why it was completely omitted from the Discussion, when some hybrid individuals were indeed detected?

Reviewer 2 Report

In the current MS, Ott et al. used an ‘integrative taxonomy’ approach to delimitate the diploid species of Leucanthemum (Compositae-Anthemideae). The MS is in good writing and I have only some minor comments:

 The ABBA-BABA test is normally used to test gene flow/introgression, though sometimes is also used to detect hybrid (i.e. Kong and Kubatko, Sys Biol, 2021), but in the current situation (only part individuals have significant D/Z value), it is better to describe as gene flow detection.  

 Abstract: It is difficult for the readers to conclude that 17 of the 20 (or 22?) Leucanthemum morpho-species are supported by genetic evidence, the results are very fragmented and do not focally concerned in the maintext.

Line 81 change ; to )

Line 83 add “, 95% credibility interval” after 2.94

Line 87 change ecology to ecological niche

Line 134, 135 rewrite the sentence with “one consisting of …, the other consisting of” or remove the “and” before L. gaudinii and L. eliasii.

Figure 1 Why not remove the 60-01 and 55-01 for the test without hybrid accessions, as both of them are with strong hybridization signal?

Figure 1 move the bar to right bottom corner

I am very confused with the L. vulgare-group in Line 137 and 192, do they indicate same or different species? And what is the relationship with L. vulgare-cluster and L. vulgare-group (i.e. Line 353 and Line 355), which should be accordant in the whole MS (the same for L. pluriflorum-cluster/group)?

Figure 3 It is reluctant to say scenario 4 has a knee point, why not scenario 5 or 9?

Table 1 why L. cacuminis, the member of L. pluriflorum group, is not included for the pairwise comparisons test?

Line 199-200 rewrite the sentence.

Table 2 Why there are different numbers for the collectors? Change et to &, or reverse.

Is there some morphological difference between the hybrid accessions 60-01, 68-04 (L. cacuminis), 55-01 (L. pluriflorum) and other without hybridization signal accessions?

Reviewer 3 Report

I enjoyed reading the manuscript and have only one non-minor comment: you are actually using an integfrative approach only in those groups, where geneological separation is weak. By ignoring other lines of evidence (ecology, geography, morphology) in species outside the vulgare- and pluriflorum-groups, you are implying that genealogical evidence is the prime source. I am not sure whether this is waht you may want to imply: as genetic data may result in oversplitting, I think that the integrative approach is also necessary for other species (even if the results will be trivial, especially for someone familiar with the group). At least it should be indicated why you don't deem it necessary to use the same approach over the entire group and only in those parts, where genetic data fail to identify clear discontinuities.

Additional comments:

l. 36: "is variable throughout" instead of "are variable throughout"

l. 81: insert closing parenthesis after "[19-22]"

l. 137: "the dataset consiting of the L." (insert "of the")

l. 138: it is not evident from Fig. 2B that datasets with and without hybrids would result in different K's

l. 163-164: re-phrase to "... and L. gallaecicum, respectively, to single groups, whuile the remaing ..."

l. 177-185: change tesne from present tense to past tense

l. 190: "...) were not significantly allopatric"

l. 264: "These two clusters" ("these" instead of "those")

l. 317: "An early formal" ("an" instead of "a")

l. 358: re-phrase as: "...MSC-based SD, in contrast to CKM clustering, showing no significant differentiation, while the ..."

l. 390: "pluriflorum" (instead of "pluriflourm")

l. 399: "population groups" (make plural)

l. 524: the background test is more appropriate for species with not or only litlle overlapping ranges

l. 607-611: the Data Availability Statement still includes the template text and has to be updated.
